# A Systems Understanding Underpins Actions at the Climate and Health Nexus

**DOI:** 10.3390/ijerph18052398

**Published:** 2021-03-01

**Authors:** Montira J. Pongsiri, Andrea M. Bassi

**Affiliations:** 1Stockholm Environment Institute, Asia Centre, Bangkok 10330, Thailand; 2KnowlEdge Srl, 21057 Olgiate Olona, Italy; andrea.bassi@ke-srl.com; 3School of Public Leadership, Stellenbosch University, Stellenbosch 7602, South Africa

**Keywords:** climate change, systems thinking, health equity, planetary health

## Abstract

Multiple sectors—health and non-health—can determine the health and well-being of people and the condition of the socio-ecological environment on which it depends. At the climate and human health nexus, a systems-based understanding of climate change and health should inform all stages of the policy process from problem conceptualization to design, implementation, and evaluation. Such an understanding should guide countries, their partners, and donors to incorporate health in strategic climate actions based on how health is affected by, and plays a role in, the dynamic interactions across economic, environmental, and societal domains. A systems-based approach to sustainable development has been widely promoted but operationalizing it for project level and policy development and implementation has not been well articulated. Such an approach is especially valuable for informing how to address climate change and health together through policy actions which can achieve multiple, mutually reinforcing goals. This commentary article describes strategic steps including the complementary use of health impact assessment, quantification of health impacts, and linking climate and health actions to national and global policy processes to apply a systems-based approach for developing climate mitigation and adaptation actions with human health benefits.

## 1. Introduction

At the current rates of warming, the world is likely to reach 1.5 °C above pre-industrial levels between 2030 and 2052 [1]. A rapidly changing climate is expected to exacerbate risks to human health [2] through direct (e.g., extreme heat), indirect, or ecosystem-mediated (e.g., malnutrition, or vector-borne disease risk), and socially-mediated (e.g., displacement) pathways [3]. This complexity of pathways makes difficult the attribution of health outcomes to climate change [4] and therefore can make actions challenging to develop and implement. The magnitude of climate-driven health impacts depends on the climate adaptation and mitigation actions taken now. Such actions must reflect a systems-based understanding of how the health and other sectors affect climate-driven health effects. Unless policy actions reflect this understanding, efforts to improve and sustain health will be undermined.

The relationship between climate change and health and well-being can be non-linear and involve time delays and feedbacks [5,6]. These complex, dynamic interactions can lead to health and well-being outcomes which are hard to predict and lead to unintended consequences. This calls for a systems approach [7,8,9] to address climate change and health together [10]. Systems thinking brings a coherent approach to inform climate change actions by requiring a consistent, rigorous evaluation for every proposed mitigation or adaptation action, including the impacts, tradeoffs of, and unintended effects on both health and climate especially on vulnerable and marginalized groups. There are several reasons why this approach adds practical value for informing needed actions on climate mitigation, adaptation and health. 

First, the use of a systemic approach can help clarify causal relationships including direct and indirect pathways between climate change and health, the strength of the relationships, which groups are differentially impacted and why, and any feedback loops which can affect these relationships in positive and negative ways. Second, this understanding helps to identify actions to prevent or reduce harm to both climate and health without ultimately undermining health due to unrecognized tradeoffs or unintended side effects. Third, a systems approach can help identify intervention points that target upstream anthropogenic drivers of health and well-being as well as their social and economic determinants rather than focus solely on the symptoms of health problems (Box 1). 

At a time when resources are limited and climate policy ambitions are flagging, country actions should aim to meet multiple objectives at the same time, so addressing health and its climate-related drivers together makes sense. For example, because of their common causes, integrated, cross-sectoral action tackling climate change and air pollution can provide significant benefits in reducing health risks in the near term at the local level [11], mitigating climate change and potentially increasing energy savings [12]. Such multiple benefits-based policies can be opportunities to maximize efficiencies and increase the effectiveness of investments to “ensure health and well-being” (United Nations Sustainable Development Goal (SDG) 3) and “take urgent action to combat climate change” (SDG 13).

Box 1Developing a systems-based understanding of climate-driven health challenges and actions to address them.Business as usual:
What are the two-way effects (positive and negative) between climate and health? What is the magnitude of the effects?What drivers of change (feedback loops) are triggered by the interlinkages existing between climate and health?Which groups are more affected and why? At what spatial scale? Over what timeframe?Do any effects influence other sectors, and if so, how? Is it possible that any vicious, negatively reinforcing cycles are triggered by the interplay of health and other sectors?
Interventions:
How does a proposed policy action affect other sectors, to what extent, at what spatial scale and over what time period? What additional feedback loops are strengthened or weakened by the intervention options? Is there a historical trend which shows a virtuous or positive reinforcing effect? Does the intervention lead to equilibrium or a balancing effect?Which groups benefit more and which groups benefit less from the desired impacts?How do non-health sector impacts then affect health and other sectors, to what extent, at what spatial scale, over what time period, and for the benefit or loss of which groups?

This commentary article describes how a systems approach to climate and health underpins the three key elements needed for effective health and non-health collaboration and actions to address the climate and health nexus: understanding multisectoral impacts of human-driven climate change, assessment of climate and health benefits and costs of policies, and skills for working cross-sectorally to share information and coordinate policy actions. The following sections describe the use of existing tools, assessment processes and strategic framing which could be used in complementary ways to apply a systems-based approach to support climate and health actions. The content is derived from selected examples and our collective country experiences but without a comprehensive literature review.

## 2. Applying a Systems-Based Approach

Climate and health actions require cross-sector collaboration through shared understanding of the impacts of climate change on health and other sectors as well as the application of this understanding to inform assessments (e.g., health impact, vulnerability, cost-effectiveness/cost–benefit) which can in turn inform strategic climate and health actions. Building shared understanding of climate change and health can be facilitated through the use of human health indicators [13]. For example, this can begin with a shared understanding of climate–health relationships involving direct and indirect pathways (including socially-mediated pathways), within a framework and methods for the identification of indicators (e.g., Causal Loop Diagrams, based on Systems Thinking [14] which cover the spectrum of problem identification, setting problem solving objectives, development, implementation, and evaluation of interventions. The *Lancet* Countdown on health and climate change [15] identifies relevant indicators updated annually across climate change impacts on health, risk exposures, vulnerability factors, adaptation, planning, and resilience; mitigation and health co-benefits; economics; and, political engagement. As has been noted, climate and health indicators that more explicitly integrate environmental sustainability and development are needed to achieve the SDG targets [16].

Systems Thinking is already used in the health sector, supporting a variety of public health issues (see for instance [17,18]). Systems Thinking is also used in other sectors, informing planning exercises in areas such as green economy, green growth, and climate adaptation [19]. Systems Thinking brings the advantage of helping to clearly identify the dynamics underlying a system, which can then help identify effective interventions for positive impact. While most of the systems thinking and system dynamics applications available to date are customized and applied in niche planning processes, this article presents the opportunity to build on and extend previous systems-based work to integrate health more explicitly in projects and policy assessments on climate and health actions; and, to use systems thinking and systems dynamics more extensively beginning with a baseline systems understanding of climate and health (Box 1). This understanding takes into account the drivers affecting the climate–health relationship including feedbacks, tradeoffs and unintended side effects across multiple sectors. Addressing these systems questions can help identify interventions to benefit climate and health while at the same time reveal other issues that should be addressed in designing solutions which can benefit all key stakeholders involved (i.e., equity). As such, this systems understanding of climate and health should underpin the use of key tools like Health Impact Assessment (HIA).

### 2.1. Mainstreaming Health Impact Assessment

Using a systems-based understanding of climate-driven health challenges as described in Box 1, HIA can be an important tool to identify, and make recommendations on, actions which benefit both climate and health. HIA is a tool which when used early in the development stage of a project or policy process helps identify likely health impacts. HIA features broad stakeholder engagement and reducing social disparities, paying special attention to the health impacts on vulnerable groups and how they may be affected [20]. In addition to quantitative data, the HIA process incorporates qualitative information which can help characterize pathways of health risk vulnerability. HIA results in evidence-based recommendations to prevent and control health risks and/or promote any health benefits. HIA can also be conducted during policy implementation and for evaluation. As the understanding of the complex and dynamic relationship between climate and health continues to grow, the prospective HIA process presents a promising opportunity for applying an updated systems-based understanding of climate and health, integrating information across disciplines and sectors, with relevant indicators in order to help identify climate-driven impacts, tradeoffs of, and unintended effects on, health. Such an understanding should inform actions needed now to address climate and health. 

For example, the HIA’s scoping stage, where relevant health impacts are identified, could set the spatial and temporal boundaries of a climate-associated health concern. For example, concerns about air pollution would consider local and urban–rural (ground-level ozone, PM_2.5_), country-wide (PM_2.5_), and transboundary (PM_2.5_) scale effects which could in turn inform potential climate mitigation actions and take into account the groups most vulnerable to exposure—children, women, the elderly, outdoor workers, those with chronic respiratory conditions, and people living in poverty [21]. Such an HIA could result in recommendations on mitigation actions which target the common cause of both climate change and short-lived air pollutants such as ground-level ozone—a missed opportunity for nearer-term realization of health benefits locally (see [11]).

HIA can identify likely health impacts of projects and inform health systems responses. HIA of the Don Sahong Hydropower Project in Lao People’s Democratic Republic (PDR) identified a greater risk of dengue due to greater availability of *Aedes aegypti* mosquito breeding sites created by land disturbance associated with the project [22]. Dengue mortality is already on the increase [23] in the Asia Pacific region, and by 2030, climate change is estimated to contribute to shifts in the geographic range of dengue-transmitting mosquitoes [24]. These geographic, spatial scale and time estimates should inform a full HIA so as to inform health systems planning, prevention and control of infectious diseases among the most vulnerable—children, pregnant women, the elderly, and those with limited access to health services [25]. 

In addition to risk assessment, HIA should be used to inform the design and scale of climate and health adaptation actions including green infrastructure-based strategies. For example, local leaders in the state of Georgia in the United States applied HIA to a green street project which was aimed at reducing stormwater runoff and flooding. The HIA resulted in the city’s decision to increase the scale of the green street project by threefold in order to capture the health benefits of walkability and reduced heat island effect [26]. 

Similarly, to reduce the intensity of urban heat islands (UHI) in megacities in the Asia region, HIA should guide the design and implementation of climate adaptation actions across different physical contexts given that UHI intensity varies by extent of economic activity, city geographic size, amount of built up area density, and season [27]. HIA could inform how interventions, such as greenspace and green walls, could be effective in reducing surface temperature [28] and importantly, the appropriately scaled site of implementation (e.g., urban street canyon, building, and neighborhood). Moreover, such strategies could be modeled to estimate UHI health impacts when implemented, singly or in combination with other sectoral approaches (e.g., energy efficiency, urban forestry, and low carbon transport).

Across the Association of Southeast Asian Nations (ASEAN) region, HIA is being used to support sustainable development based on the recognition that health is a cross-cutting issue amidst rapid economic integration. HIA has been included in the Environmental Impact Assessment of projects in Lao PDR, Cambodia and Malaysia. HIA has been codified in national governance [29] and is a shared priority for regional action with progress regularly reported [30]. In the environmental health context, Thailand is the lead ASEAN country focal point on HIA. Through the development of a HIA practitioners’ training manual, Thailand is developing and promoting HIA to strengthen capacity at country level across the region. To this end, the ASEAN Environmental Health and HIA Network was established to share performance indicators and best practice [31]. The next phase of HIA work in the region is to implement improved knowledge sharing mechanisms (e.g., curricula) to reach the broad range of HIA stakeholders—the public and private sectors, academia, and civil society.

### 2.2. Quantification of Health Impacts

Quantified health impacts can facilitate collaboration between health and other sectors on climate and health. A healthy human capital base is fundamental to sustaining economic growth and development [32]. Improved understanding of health impacts, positive and negative, of climate policies is necessary to include in decision making on strategies to achieve climate and health goals. A systems-based understanding of climate-health relationships and HIA can inform the estimation of health impacts and associated costs calculated in terms of health care treatment, lost work productivity, and GDP [33] Quantification of the costs and benefits to health associated with action and inaction can support priority-setting and selecting among climate-health actions under consideration. Health risks that are identified could be managed proactively through preventive measures which include increasing awareness of climate change–health relationships among the health workforce; improved health information systems for integrated risk monitoring and early warning; and greater financing [34]. Quantification of health and environmental impacts can support policymaking by providing economic bottom line estimates for costing of climate actions and informing cost-effectiveness analyses of policies for achieving climate and health objectives. In fact, the quantification of health impacts can serve two purposes: it highlights the economic cost of side effects, originating from policies and projects that do not take into account health outcomes; and, it highlights the extent to which climate adaptation and mitigation projects can reduce and avoid health costs. The quantification of these two impacts and related economic valuation can inform decision making by estimating the net societal value of a given project, equally weighing benefits and avoided costs. This will make it more difficult, and certainly undesirable, to ignore the role of human health in project conceptualization and policy formulation. The potential health benefits of climate actions as expressed in avoided/reduced costs to economic and social development could make policy actions and projects more financially viable to investors, or attract different investors, those vulnerable to the impacts often unaccounted for, in both the private and public sectors.

Quantification of climate and health effects depends on the strength of their causal relationship. While attribution can be complicated, there are climate and health relationships that have been studied with high confidence [see3]. Quantified health effects could significantly increase visibility of climate change–health issues for the finance sector as well as for the public interest. This could be key to facilitating greater cooperation between health, environment, finance and other sectors. 

Several analyses have shown the economic rationale for taking climate actions including for health cost savings. Climate mitigation efforts to meet the Paris Agreement in transport and power generation which reduce air pollution could exceed the costs of implementation by a factor of 1.4 to 2.45 in China and India, respectively, with savings of trillions of dollars in worker productivity and health care expenses globally [35]. Interventions of transport mode shifts which facilitate physical activity and healthier lifestyles could lead to even greater cost savings associated with noncommunicable diseases [36]. An evaluation of a market based regulatory program to reduce greenhouse gas emissions at the subregional level in the United States estimated cost savings of avoided adverse children’s health outcomes related to annual average secondary PM_2.5_ reduction in the range of $191 to $350 million between 2009 and 2014 [37]. Reducing deforestation can have potential health as well as climate mitigation benefits. It can limit the emergence of zoonotic diseases such as severe acute respiratory syndrome (SARS) associated coronaviruses like SARS-CoV-2. Such measures can be taken at a cost that is significant orders of magnitude less than the costs that COVID-19 disease has already imposed in morbidity, mortality, and loss of GDP while producing social benefits from reduced greenhouse gas emissions [38].

There are many methodologies for estimating the health benefits of actions taken in non-health sectors, and they may also include associated economic cost savings and damages [39,40,41]. However, there are important differences when quantifying the health benefits of climate mitigation or adaptation which can complicate generalizations about health impacts and associated economic costs. Notable considerations in estimated quantified costs associated with health impacts include methodological differences in the type of health outcome measured; how the health outcome is quantified; assumptions made about the risk exposure—health response function; uncertainties about confounding factors; the spatial and temporal scales considered; and, the specific policy scenarios modeled [42]. 

### 2.3. Capitalizing on Efficiencies by Linking Climate–Health Actions in Support of other National and Global Policy Processes

Together with the use of health indicators, HIA, and health quantification, strategic framing of climate and health can further support how a systems-based approach can be applied to climate and health actions. The SDGs are a strong global framework for identifying strategies which exploit synergies between climate change and health related SDGs [43]. By taking advantage of countries’ flexibility in the means of implementation to achieve the SDGs, there are opportunities to build on the SDGs commitment to identify health and non-health sector strategies to reach targets in an integrated way [44]—specifically, improving health by addressing its upstream human-driven environmental drivers. For example, using clean energy for household cooking is a good public health intervention to reduce indoor air pollution (SDG 3) while providing access to other forms of energy (SDG 7). 

There is a gap between recognition of climate–health relationships and policy action to address them. In a review on health in the Nationally Determined Contributions (NDCs) [45], 86 of the assessed 184 countries recognized the impacts of climate change on health (especially vector- and water-borne diseases as well as food security), but only 10 countries out of 184 included evidence or policies on climate change’s impacts on health in their NDCs, and only 3% of NDCs highlighted the health benefits of climate actions. 

When resources are limited, policy actions should aim to meet multiple objectives especially when the objectives are mutually reinforcing. In addition to the SDGs and the Paris Agreement, other country level commitments or frameworks should be built upon to maximize efficiencies in achieving climate and health goals at the same time. These include the Health in All Policies (HiAP) country framework and Universal Health Coverage (UHC). With the premise that health is affected by all public policies, the Health in All Policies (HiAP) country framework highlights multisectoral collaboration to improve health and health equity [46]. Strategic coordination of health and other sectors affecting health could result in actions which achieve efficiencies in energy use, climate change, and health together, with significant cost savings [47]. Our proposed systems-based approach to climate and health involves the health sector working with other sectors which influence health. Integrating health, climate change and social determinants of health is necessary to achieve health goals in a more holistic and equitable way by targeting environmental drivers and social determinants so as not to ultimately undermine health. Collaboration between health and other sectors can be facilitated by the HIA tool and HiAP framework, underpinned by a systems-based understanding which should appreciate the depth of knowledge and experience of partners working in non-health sectors. Capacity development among health and non-health sectors on climate–health understanding and methods for cross-sector engagement should be strengthened. 

Given the urgency to act on climate change, especially with a health lens, policymakers should take advantage of HiAP’s principles of cross sector collaboration on country-led climate actions. A systems approach brings coherence to building understanding of climate change and health relationships and applying that understanding to policy actions which benefit both climate and health while minimizing adverse tradeoffs and unintended side effects including disproportionate impacts on vulnerable groups which may have gone unrecognized without such an approach. At a minimum, the systems approach can help to highlight these issues so that they can be addressed early.

Achieving UHC is one of the primary policy goals of the World Health Organization [48]. However, many regions of the world with the highest vulnerability to climate change are also those with the lowest levels of UHC coverage [49]. UHC cannot be achieved without addressing climate change because population health is affected by climate change, and the health system itself contributes to climate change, responsible for 4.4% of global carbon emissions [50]. Climate-driven health outcomes should be included in the essential health services coverage by way of workforce training on climate–health relationships, financing, and increasing resilience of health care service delivery which may be disrupted during climate related events (e.g., storms, and flooding). These can bolster UHC to more effectively address context-specific climate driven health effects which are already being experienced and which are expected to worsen over time. In addition to the project-level HIA, the National Health Adaptation Plan (NHAP) (see [13]) should be developed and used to directly inform climate adaptation policies. The NHAP is intended to integrate health in adaptation planning by identifying vulnerabilities in the health system as well as opportunities to increase the resilience of health systems to climate change. Having in place an NHAP is an indicator of a country’s readiness to identify and respond to climate-associated health events (see [23]). The World Health Organization identified vulnerability and adaptation assessment as one of the key components of a country level operational framework for climate resilient health systems (see [34]). In the climate and health context, a systems-based HIA can take the form of vulnerability and adaptation assessment [51] that is designed to build a baseline understanding of climate and health (Box 1) so as to use that understanding to inform interventions to reduce climate-driven health risks. In addition to assessing current vulnerabilities and impacts on health, the systems-based HIA process should consider the vulnerability of other sectors (e.g., food) and the health impacts of that vulnerability [ibid].

HIA, updated to incorporate a systems-based understanding of climate and health can address risk factors of vulnerable groups as well as the capacity of the people, institutions and resources of existing health systems to prepare for and adapt to climate-driven health challenges. Because NHAP and HIA both prioritize cross-sector cooperation and reducing health inequities, they are consistent with, and therefore should be used to advance the implementation of, the country’s HiAP goals.

### 2.4. Implications for Decision Support

A systems-based understanding of climate change and health—i.e., the human drivers affecting their relationship including feedbacks, tradeoffs, and unintended side effects across societal, environmental, and economic domains—can help identify interventions to benefit climate and health while at the same time reveal other issues that should be addressed in designing solutions to benefit all key stakeholders involved. 

For example in 2015, approximately 100,000 deaths across Singapore, Malaysia, and Indonesia were attributed to fires which were set to clear peatlands for palm oil production (Figure 1). 2015 was an El Nino year which created drought conditions, and with the drier conditions, peat provided abundant fuel. The fires burned more intensely and for a longer period of time, releasing significant carbon emissions as well as harmful health pollutants such as fine particulate matter which moved downwind towards population centers [52,53]. The estimated adult mortality (and health care costs) associated with specific fire events justify the need for peatlands protection and enforcement of fire prevention. To date, combined efforts to restore peatlands and rewet degraded peatlands have been missed opportunities to prevent the further release of locked-up carbon and to enhance the function of peatlands as important carbon sinks, contributing to climate mitigation efforts globally [54]. Notably, the benefits and costs of peatland protection are not equally borne. Farmers’ livelihoods must be considered [55] as part of such potential nature-based solutions possibly through payment for ecosystem services, subsidies, and other incentives.

Legend: As described by Sterman [56], causal loop diagrams (CLDs) include variables and arrows (called causal links), with the latter linking the variables together with a sign (either + or −) on each link, indicating a positive or negative causal relation (see Box 1). A causal link from variable A to variable B is positive if a change in A produces a change in B in the same direction. A causal link from variable A to variable B is negative if a change in A produces a change in B in the opposite direction. Circular causal relations between variables form causal, or feedback, loops. There are two types of feedback loops: reinforcing (R) and balancing (B). The former can be found when an intervention in the system triggers other changes that amplify the effect of that intervention, thus reinforcing it [57]. The latter, balancing loops, tend towards a goal or equilibrium, balancing the forces in the system [ibid]. 

Nature-based solutions are an emerging area of research which has the potential to both improve health and reduce greenhouse gas emissions. For example, forests not only support water filtration [58] but also provide wild foods which contribute to dietary diversity [59,60]. Conserving forest cover in close proximity to communities was associated with approximately 7% reduced stunting in young children across 25 low- and middle-income countries, comparable to the median effect of traditional interventions such as providing micronutrient supplements and food fortification [61]. The quantified potential of forest conservation as a public health intervention in rural areas could capitalize on high population coverage due to the large numbers of nutrition insecure groups who live there, suggesting that health nutritionists and conservation practitioners could collaborate to include protecting forest cover as part of the toolbox of interventions to improve the health of vulnerable populations while also maintaining forests’ carbon sink role. 

In addition to identifying interventions to address the climate-health nexus, a systems-based understanding can help to highlight disproportionate health impacts on vulnerable populations, amid the variation in types of predominant health impacts of climate change within and across countries and regions [62]. Within a given geography, climate-driven health effects will vary by equity aspects related to age, gender, income, livelihoods, and capacity to successfully address them [63]. Projects must be responsive to community needs, and climate actions must be context-specific for there to be successful implementation. These can be achieved through an inclusive approach to building a systems-based understanding of climate change and health to help identify disproportionately affected groups and how they could be impacted over time. For example, a systemic approach applied to a large infrastructure investment for hydropower in Cambodia revealed that local populations in the northern provinces of the country would be affected both positively and negatively by the project. Positive outcomes included improved access to education and health, via the construction of roads and new facilities. On the other hand, there were estimates of reduced access to resources, such as water and fish, possibly leading to poorer nutrition and negative impact on livelihoods [64,65]. 

### 2.5. Financing Climate and Health Actions

The relationship between climate and health is not trivial, especially for investors. Climate change cannot be predicted with certainty, which reduces the interest of private investors in climate adaptation due to the lack of a revenue stream or a clear trend of avoided costs. On the other hand, the cost of climate adaptation is too high for most governments which already struggle to maintain operational existing infrastructure, especially in low and middle-income countries [66]. As a result, in order to invest in an area with high financial risk, but with the potential to generate benefits across a multitude of economic actors, a new approach to financing is required. Such an approach has to combine the “high risk-high reward” perspective of private investors with the need to provide basic services (often not economically viable) by governments and donors. 

There are innovative financing tools which can complement public funds and public sector strategies (e.g., taxes, and loans) to address climate and health actions. For example, parametric or index-based insurance instruments can support climate adaptation in the health sector by reducing the impacts of extreme weather events on health and also increasing funding for health-related assets. This type of insurance monetizes climate risks and therefore can increase the financial viability of investments aimed at reducing these risks. Index-based insurance offers payouts even when there is no physical damage, when a pre-determined set of parameters such as weather conditions of a certain type and severity occur [67]. Payouts are made relatively quickly, as compared with traditional insurance, to fund emergency responses as well as for investments to improve climate resilience. For example, to address food security regularly threatened by drought, Mauritania was one of the earliest purchasers of parametric insurance through which it was given an estimated premium of US$ 1,394,000 for a total guaranteed cover of US$ 9,000,000 for the agricultural season from July through November 2014. After the next drought, Mauritania was eligible for a payment of approximately US$ 6,326,000 which was paid out in January 2015. The Mauritanian government used the quickly distributed funds to reduce the impact of drought and to protect livelihoods by providing 50,000 vulnerable households with 50 kg of rice and 4 L of oil each over 4 months. These measures also reduced stress-driven migration and the distressed sale of livestock [ibid]. In this way, parametric insurance could be an important tool to incentivize resilience planning as part of a proactive preventive approach to health system preparedness for climate adaptation especially in high-risk areas. The tool is flexible enough to adapt to local needs and context by modifying the specific climate parameters that trigger payment. 

## 3. Discussion

A systems-based understanding of climate change and health is critical across all stages of the policy or project process—from problem conceptualization, design to implementation, and evaluation. This has important implications for the development of policies as well as project proposals aimed at the climate and health nexus.

Achieving a shared understanding on the central role that human health plays in dynamics triggered by climate change and their socio/economic consequences is not trivial. We are surrounded by complexity, driven by the growing interconnection of social, economic and environmental dynamics of change. Regardless of how challenging it may be, our capacity to plan must adapt to such growing complexity if we want to maximize the societal value of investments, increase economic resilience and achieve national and global development targets (e.g., SDGs). We have the knowledge and tools to tackle such complexity. Communication and knowledge integration are of critical importance. We have developed great depth of knowledge on health and climate change, and in many other related fields. Experts must come together, using a multi-disciplinary and multi-stakeholder approach, because no one has all of the knowledge required, and communicate about the critical interrelationships between climate and health. When this integration takes place, especially in the context of project development and policy formulation, the complexity can be greatly reduced.

Systems thinking is a method that aims to simplify complexity and bring a coherent approach to inform climate change actions by requiring a consistent, rigorous evaluation for every proposed mitigation or adaptation action, including the impacts, tradeoffs of, and unintended effects on, both health and climate especially on vulnerable and marginalized groups. Box 1 summarized how systems thinking can support understanding of climate change–health relationships and identifying actions based on that understanding. A systems-based approach brings practical value to addressing the climate–health nexus because it can help clarify causal relationships, the strength of the relationships, which groups are differently impacted and why, and any feedback loops which can affect these relationships in positive and negative ways. Importantly, for identifying intervention actions, a systems-based approach can help to prevent or reduce harm to both climate and health without undermining health due to unrecognized tradeoffs or unintended side effects. In the COVID-19 recovery period, the use of relatively limited resources should be maximized to achieve interrelated economic, environmental and health goals concurrently. Actions which benefit both climate and health are opportunities to maximize efficiencies, by reducing costs and generating new benefits. This increases the financial viability of investments, especially from a societal perspective, where a single investment may result in several beneficiaries across sectors, and over time. To apply systems thinking for climate and health actions, cross-sector collaboration between health and non-health sectors is a key element. This can be facilitated through the complementary use of existing tools, assessment processes and strategic framing of climate and health actions to meet related country and global commitments.

Within countries charged with developing climate actions as well as the organizations (e.g., funders, and accredited entities) which support them, the systems-based approach must be the common starting point. Donors should require a systems-based assessment in project proposals. Specifically, funders such as The Green Climate Fund (GCF) should require health in all climate mitigation and adaptation projects, making clear how health plays a role in the dynamic interplay across economic, environmental and societal domains. 

Donors, bilateral and multilateral development banks, as well as governments should mandate the use of HIA. For country climate adaptation and mitigation project developers, HIA should be used as a good governance tool. As the scientific study on climate change and health continues to advance, spatial scale and time relevant climate–health inter-relationships, feedbacks, and tradeoffs among health and non-health sectors need to be reflected in the HIA. This baseline understanding should then inform potential climate mitigation/adaptation interventions which minimize adverse effects especially on vulnerable or marginalized groups. 

Further research is necessary to advance a systems-based approach to inform actions at the climate–health nexus. Practically, new research is required every time a systems approach is used because any given context is unique if we consider how social, economic, and environmental drivers of change interact with one another. Further, research is needed on specific topics to generate the evidence base that can be further tested, customized and validated during project conceptualization and policy formulation. For instance, we need an improved understanding of how climate change interacts with other environmental changes to affect health, given the potential for negative synergistic interactions [68]. Nature-based solutions is an area that is ripe for study [69] particularly for evaluating and quantifying the potential of actions to meet health objectives [70]. While research studies may not have been originally designed to demonstrate both health and climate benefits, their findings could inform potential nature-based solutions which could be implemented with field partners at the appropriate spatial scale and evaluated through implementation research [71]. Simulation models could be used to forecast outcomes across sectors and for many economic actors, strengthening development planning (see [19]). Selection of targets and indicators relevant to health are key considerations for designing and measuring the effectiveness of climate mitigation and adaptation (see [44]). Moreover, HIA recommendations should focus on the upstream human-driven environmental and social determinants of health so as to avoid the adverse effects to be borne by the most vulnerable groups and the costs of health externalities to be transferred to the health sector [72]. The mandatory use of HIA early in the climate mitigation/adaptation project development process can socialize the viability and sustainability of projects which include appropriate and measurable health indicators and avoided/reduced costs to economic and social development by recognizing health (see [22]). 

From a more practical perspective, donors including the Green Climate Fund should request that project developers prepare a climate and health cost–benefit assessment for every proposed project. This is critical, especially for low and middle-income countries, to create value for money for public investments (see [66]). Integrating health in all projects and in all policies will allow for maximizing the performance of several sectors and indicators of performance, simultaneously, as decisionmakers could recognize that their decisions may lead to extra health costs. This can be achieved by avoiding the emergence of future costs (e.g., via increased health care-related infrastructure resilience to climate change), and by creating additional benefits (e.g., via increased labor productivity). Practically, value for money for every investment improves if health is explicitly considered in the project planning stage.

In this article, we propose a systems-based understanding as the basis for a coherent approach to climate and health—specifically, by using health impact assessment, quantification of climate-driven health impacts, and strategic framing in a complementary way to support climate and health actions. The content of this commentary article is derived from selected examples and our collective country experiences but without a comprehensive literature review. There is a need for more case studies which build on available systems-based applications in health and other sectors which affect health. In addition, an extensive review of the literature may be helpful for providing more examples of how the HIA tool and HiAP framework could be used to apply the proposed systems-based understanding and to inform specific guidelines on actions which benefit climate and health.

## 4. Conclusions

The health and environment sectors are impacted by policies made across sectors which can determine the health and well-being of people and the socio-ecological system on which it depends. A systems-based approach to climate and health can be operationalized by first understanding the systemic two-way health and well-being impacts of human-driven climate change; feedback loops triggered by climate–health relationships; how and why there are disparities in health and well-being impacts; and, which groups are the most vulnerable. This paper describes the value of, and how to apply, the systems-based approach to address the climate and health nexus. This does not begin with a blank slate. Already existing tools such as HIA can be used strategically to inform climate and health understanding and actions while also advancing the SDGs and prominent country goals such as Health in All Policies (HiAP), and strengthening health systems for Universal Health Coverage (UHC). Donors and project/country partners can play important roles to incentivize and support the development of science-based actions which reflect systems thinking. Quantified health impacts should be used in cost-effectiveness or cost–benefit assessments of policies, supporting the financial viability of climate actions and as a result attracting traditional and innovative financing instruments. At the same time, the strategic consideration of health in climate mitigation and adaptation strategies must be enabled by governance structures and the capacity to work cross-sectorally, as part of a continuing effort to improve the resilience of health systems. The strategic, practical steps described in this paper should serve as 2021 United Nations Climate Change Conference (COP-26) entry points for a more concrete process to take climate and health action.

## Figures and Tables

**Figure 1 ijerph-18-02398-f001:**
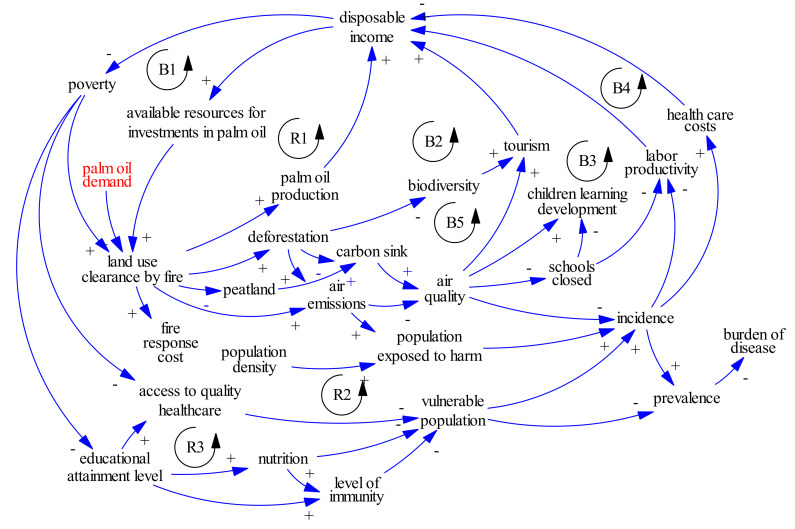
Systems diagram representing the dynamics at play in relation to peat fires.

## Data Availability

Not applicable.

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
