# Peer review of "A Systems Understanding Underpins Actions at the Climate and Health Nexus"

_ijerph, 2021, doi:10.3390/ijerph18052398_

Round 1

Reviewer 1 Report

Reviewer comments on “A systems understanding underpins actions at the climate and health nexus” by Montira J. Pongsiri and Andrea M. Bassi, submitted to the International Journal of Environmental Research and Public Health

Overall comments:

This important article should serve as a call to action for scientists, resource managers and policymakers to simultaneously consider environment and human health outcomes when devising, optimizing and implementing solutions to inter-linked human health, climate and environment challenges. The examples given demonstrate the value of this holistic perspective and provide useful information for those who can – and should – consider an integrated human health and environment perspective when evaluating project design and developing policy.

In light of the importance of the holistic approach described and the stakeholders who need this information, I recommend the article be adjusted to more easily reach a broader audience. For example, I recommend reducing use of jargon and reformulating the abstract to emphasize take-home messages. In addition, because those coming from the environment sector may use the term “environmental health,” the article should be clear that “health” here appears to always refer to human health.

To fortify their perspective, a paragraph the authors may wish to add in the Discussion is a perspective regarding what potential critics of a holistic, “systems approach” may believe. For example, a detractor may assert ‘It’s too complicated to manage human health and environment outcomes at the same time’ or ‘We must remain focused on human health but not on other complex, global issues like climate change that are out of our control.’ This article presents an opportunity to convince policymakers about the necessity and utility of a systems approach.

I am curious if the authors are dialed into the American Geophysical Union’s GeoHealth effort. There may be useful touchpoints there. For example, see https://www.agu.org/Share-and-Advocate/Share/Policymakers/Position-Statements/Fact-sheet-Geohealth.

Detailed comments and edits:

Lines 10-16 (abstract): The abstract lacks punch and includes awkward phrases and jargon, which will limit readership of the article. I suggest a rewrite in simpler language oriented to a lay audience. For example, the first sentence is long and contains “sectors” twice, which is confusing. The first sentence uses “system” in a way that is different than “systems-based approach” in the second sentence. And “systems-based approach” – essential to this article – isn’t really defined until line 78. Last, the urgency of the opportunity described in the article isn’t emphasized in the abstract but should be.

Line 24: Repeated use of a word in the same sentence: “This complexitycomplicates attribution…”

Line 27-28: Awkward phrasing in “…must reflect a systems-based understanding of how health as well as non-health/health influencing sectors…” Best to deconvolute the language so as to be more clear for the reader.

Line 36: The background and thesis of the article needs to be more firmly developed before sending the reader to Panel 1. I suggest the reference to Panel 1 be moved to a later paragraph.

Line 41 (in Panel 1): Why is “baseline” equated with “scenario of inaction”? The baseline includes all current actions, not a lack of action. I think the authors mean “business as usual” instead of “baseline” here? Regardless, this should be made more clear.

Lines 68-76: The paragraph that begins on Line 68 makes the primary assertion of the article: to address climate and health policy needs simultaneously. This key point should be included in the abstract for those who only read the paper’s summary.

Line 70: I suggest removing “to do”.

Line 75: I suggest adding “United Nations” prior to “Sustainable Development Goal,” at least at first use. The acronym “SDG” can be parenthetically defined here as well instead of later.

Lines 78-81: I was waiting for a definition of “systems thinking” as it pertains to this article and see it here. For clarity for the reader, this definition should be moved earlier in the article.

Line 87: I tripped up again on “health and non-health/health influencing sectors.” Can this be written more simply?

Lines 87-88: To clarify: are the authors asserting that “health indicators” are the only window into monitoring human health and environment outcomes, or are they just one example? And does “health” here only refer to human health, or also environmental health? (I believe just the former, which raises the question of whether environmental health and financial indicators are sometimes appropriate or even occasionally more insightful.) Either this paragraph needs a broader first sentence, or the article needs to mention early on that “health indicators” are the key mechanism around which to design effective projects and policy relevant to both the human health and environment sectors.

Line 104-105: Are “broad stakeholder engagement” and “multisectorial consideration” redundant? (I favor the former, and worry “multisectorial consideration” would be impenetrable for some readers.)

Line 119: Should “PDR” be spelled out at first use?

Line 136: I suggest changing “could” to “should” so the point becomes a stronger assertion of the authors’ position. “Could” is used in other places in the article as well, where I again suggest stronger language.

Line 178: I believe the authors meant to say “…by a factor of 1.4 to 2.45…”

Lines 186-191: This is a long sentence with important nuggets and should be broken into smaller pieces. On line 187, I suggest changing “prevent” to “limit”. (“Prevent” is a tall order.)

Line 193-198: The authors make an important point, but do not seem to provide a path forward. What are we to do with the “important differences” noted on line 196? The authors should expand here.

Line 209: I suggest changing “modern energy” to “other forms of energy” so as to avoid suggesting that current, widespread approaches to cooking are not somehow ‘modern.’ (Everything occurring today is “modern” by definition.)

Line 294: What is the difference between “health” versus “well-being” impacts?

Line 296: What is “ability to cope”?

Lines 341-343: I wonder if making an analogy to environmental impact statements would be useful here.

Lines 334-394: The Discussion and Conclusions section are clear and strong. I recommend that more of this language be brought into the Abstract and Introduction because many readers won’t make it this far in the paper.

Author Response

Thank you very much for reviewing the manuscript and providing constructive comments. Please see our responses (in bold) which are presented below the specific comments received (copied here for reference)

Reviewer 1

Overall comments:

This important article should serve as a call to action for scientists, resource managers and policymakers to simultaneously consider environment and human health outcomes when devising, optimizing and implementing solutions to inter-linked human health, climate and environment challenges. The examples given demonstrate the value of this holistic perspective and provide useful information for those who can – and should – consider an integrated human health and environment perspective when evaluating project design and developing policy.

In light of the importance of the holistic approach described and the stakeholders who need this information, I recommend the article be adjusted to more easily reach a broader audience. For example, I recommend reducing use of jargon and reformulating the abstract to emphasize take-home messages. In addition, because those coming from the environment sector may use the term “environmental health,” the article should be clear that “health” here appears to always refer to human health.

We have reviewed the article to reduce jargon language. The abstract has been revised to include key take away messages. The first references to “health” in the abstract (line 13) and introduction (line 31) sections are clarified to refer to human health.

To fortify their perspective, a paragraph the authors may wish to add in the Discussion is a perspective regarding what potential critics of a holistic, “systems approach” may believe. For example, a detractor may assert ‘It’s too complicated to manage human health and environment outcomes at the same time’ or ‘We must remain focused on human health but not on other complex, global issues like climate change that are out of our control.’ This article presents an opportunity to convince policymakers about the necessity and utility of a systems approach.

We have added a paragraph at the beginning of the discussion section (beginning line 456) describing that it is critical to use a systemic approach in order to realize development targets (i.e. reality is becoming more complex, we have to adapt to such complexity and improve our approach to planning) regardless of how difficult this may be. Further, we have indicated that there are methods and tools that “simplify” complexity, based primarily on knowledge integration, for which a multi-stakeholder approach is required.

I am curious if the authors are dialed into the American Geophysical Union’s GeoHealth effort. There may be useful touchpoints there. For example, see https://www.agu.org/Share-and-Advocate/Share/Policymakers/Position-Statements/Fact-sheet-Geohealth.

Thank you for pointing to the AGU resource. In addition to research-based resources and similar efforts such as the Future Earth Health Knowledge Action Network (https://futureearth.org/2019/07/12/the-future-earth-health-knowledge-action-network-is-launched/), we intended this article to focus more on how to help translate climate-health research to policy action and projects which could advance adaptation and mitigation actions. This article should be considered a commentary article rather than a research article.

Detailed comments and edits:

  1. Lines 10-16 (abstract): The abstract lacks punch and includes awkward phrases and jargon, which will limit readership of the article. I suggest a rewrite in simpler language oriented to a lay audience. For example, the first sentence is long and contains “sectors” twice, which is confusing. The first sentence uses “system” in a way that is different than “systems-based approach” in the second sentence. And “systems-based approach” – essential to this article – isn’t really defined until line 78. Last, the urgency of the opportunity described in the article isn’t emphasized in the abstract but should be.

The abstract has been revised accordingly.

  1. Line 24: Repeated use of a word in the same sentence: “This complexitycomplicatesattribution…”

This has been revised in line numbers 33-34.

  1. Line 27-28: Awkward phrasing in “…must reflect a systems-based understanding of how health as well as non-health/health influencing sectors…” Best to deconvolute the language so as to be more clear for the reader.

This has been revised to be clearer in lines 37-38.

  1. Line 36: The background and thesis of the article needs to be more firmly developed before sending the reader to Panel 1. I suggest the reference to Panel 1 be moved to a later paragraph.

Panel 1 has been moved further down to lines 96-114 so that it is preceded by the explanation of why a systems-based approach adds practical value for informing needed actions on climate mitigation, adaptation and human health.

  1. Line 41 (in Panel 1): Why is “baseline” equated with “scenario of inaction”? The baseline includes all current actions, not a lack of action. I think the authors mean “business as usual” instead of “baseline” here? Regardless, this should be made more clear.

This has been revised accordingly in line 98.

  1. Lines 68-76: The paragraph that begins on Line 68 makes the primary assertion of the article: to address climate and health policy needs simultaneously. This key point should be included in the abstract for those who only read the paper’s summary.

This assertion has been included in the abstract in lines 20-21.

  1. Line 70: I suggest removing “to do”.

 This has been removed in line 83.

  1. Line 75: I suggest adding “United Nations” prior to “Sustainable Development Goal,” at least at first use. The acronym “SDG” can be parenthetically defined here as well instead of later.

This has been revised in line 88, and “Sustainable Development Goals” subsequently referred to as “SDGs”.

  1. Lines 78-81: I was waiting for a definition of “systems thinking” as it pertains to this article and see it here. For clarity for the reader, this definition should be moved earlier in the article.

This has been moved earlier to Lines 46-49.

  1. Line 87: I tripped up again on “health and non-health/health influencing sectors.” Can this be written more simply?

This has been simplified to “health and other” or “health and non-health” in lines 37-38 and 116, respectively, in the context of cross-sector collaboration on climate and health.

  1. Lines 87-88: To clarify: are the authors asserting that “health indicators” are the onlywindow into monitoring human health and environment outcomes, or are they just one example? And does “health” here only refer to human health, or also environmental health? (I believe just the former, which raises the question of whether environmental health and financial indicators are sometimes appropriate or even occasionally more insightful.) Either this paragraph needs a broader first sentence, or the article needs to mention early on that “health indicators” are the key mechanism around which to design effective projects and policy relevant to both the human health and environment sectors.

This has been revised to describe that the use of health indicators is just one example for monitoring climate-health relationships (line 130). “Health” has been clarified to refer to human health (line 129).

  1. Line 104-105: Are “broad stakeholder engagement” and “multisectorial consideration” redundant? (I favor the former, and worry “multisectorial consideration” would be impenetrable for some readers.)

This has been revised by removing “multisectoral consideration”.

  1. Line 119: Should “PDR” be spelled out at first use?

This is spelled out in line 172.

  1. Line 136: I suggest changing “could” to “should” so the point becomes a stronger assertion of the authors’ position. “Could” is used in other places in the article as well, where I again suggest stronger language.

This is revised as suggested in line 182. “could” has been replaced by “should” in Lines 190, 480, 498.

  1. Line 178: I believe the authors meant to say “…by a factor of4 to 2.45…”

This is revised in line 255.

  1. Lines 186-191: This is a long sentence with important nuggets and should be broken into smaller pieces. On line 187, I suggest changing “prevent” to “limit”. (“Prevent” is a tall order.)

This has been condensed into separate sentences (lines 264-269), and “limit” has replaced “prevent” in line 265.

  1. Line 193-198: The authors make an important point, but do not seem to provid.e a path forward. What are we to do with the “important differences” noted on line 196? The authors should expand here.

Differences in quantification methods have been included in lines 276-280.

  1. Line 209: I suggest changing “modern energy” to “other forms of energy” so as to avoid suggesting that current, widespread approaches to cooking are not somehow ‘modern.’ (Everything occurring today is “modern” by definition.)

This has been revised in line 293.

  1. Line 294: What is the difference between “health” versus “well-being” impacts?

“Well-being” has been eliminated since “health” is inclusive (line 400).

  1. Line 296: What is “ability to cope”?

The ability to cope refers to the capacity to successfully deal with climate driven health risks and impacts. This has been clarified in line 403.

  1. Lines 341-343: I wonder if making an analogy to environmental impact statements would be useful here.

Respectfully, we would like to keep the focus on health impact assessment as a critical tool for linked climate-health actions.

  1. Lines 334-394: The Discussion and Conclusions section are clear and strong. I recommend that more of this language be brought into the Abstract and Introduction because many readers won’t make it this far in the paper.

The abstract has been revised in lines 13-17, 20-21.

Reviewer 2 Report

I agreed to review this manuscript because the topic sounded promising. While the manuscript is well written grammatically, the content did not meet my expectations. 

There was very little new knowledge presented in this contribution and the examples given focussed on air pollution / respiratory health (expect for the forest example and nutrition) - in general, the examples seemed to be cherry-picked or based on authors' knowledge rather than from an extensive review of the literature.

The article is not 'new research'; it is commentary proposing the use of systems thinking for health and climate adaptation. This has already been discussed in the literature in various contexts (see for example https://www.nature.com/articles/s41558-018-0102-4; https://www.pnas.org/content/116/17/8214; https://publichealthreviews.biomedcentral.com/track/pdf/10.1186/s40985-016-0032-5.pdf) yet none of these previous studies were mentioned.

What was disappointing was that the exact way that this 'new' approach could be implemented was not explained in sufficient detail for one to actually do it. Many statements made in the manuscript are sweeping and without substantiation or justification. For example, lines 157 - 159 what exactly does 'a fuller accounting of health impacts .... of climate policies is necessary.....' mean? Line 162, how can one manage proactively? Line 172, how can one value environmental health services? or identify the cost of actions needed for health systems to prevent, manage and respond...'? The reader is left to figure this out for themselves. Another example of the vagueness recurring in the manuscript is in line 196 - what are the important differences mentioned here? They are stated as being important yet are not elaborated upon. Line 322, how does one reduce the impacts of extreme weather on health? There are numerous examples in the manuscript where the reader is left wondering what the authors mean.

The peat fire and palm oil example is interesting, but the specifics of the example make it difficult to see how this would be done for other climate-health problems. A more generic framework could help the reader 'see' what they would need to do to address any pressing problem.

The Discussion section introduces text relating to the Green Climate Fund and seems to give the Fund specific guidance. Why is only this Fund mentioned? Other funds and donors support climate change and health adaptation. This was an odd discussion that did not bring the content of the manuscript together, instead it dealt with a different topic. This made the reader even more confused. 

Specific comments

The introduction should cover other articles that that considered systems thinking in climate change (and health) impacts as well as define what this manuscript sets out to achieve and how. There is no mention of methods - the authors should at least state this is an opinion piece and that the content is derived from select examples without a full review of the literature. In fact, such a review of the literature may be useful to provide more extensive examples of how the proposed systems-based understanding of climate change and health (using HIA and HiAP) and to then develop a framework or guidelines on exactly how to make use of this approach.

A short methods section is needed - either as standalone or as part of the introduction.

Please explain or given examples for the term 'non-health/health influencing sectors' at first use.

Line 345, what is Readiness resources? A reference is needed if this is a specific product.

Paragraph ending on line 292 does not relate well to the paragraph starting on line 294 - please create better flow.

Please check the manuscript for very long sentences - sometimes five lines long - that make comprehension difficult.

Line 317 - why the focus on private investors? Funds can come from elsewhere.

Line 324 - please give concrete examples of the insurance being discussed here and where it is being applied in the world (preferably not another Asian example).

In several paragraphs, there are no references. The authors should carefully check each sentence and make certain that the references are included to substantiate points made.

Author Response

Thank you very much for reviewing the manuscript and providing constructive comments. Please see our responses (in bold) which are presented below the specific comments received (copied here for reference)

I agreed to review this manuscript because the topic sounded promising. While the manuscript is well written grammatically, the content did not meet my expectations. 

There was very little new knowledge presented in this contribution and the examples given focussed on air pollution / respiratory health (expect for the forest example and nutrition) - in general, the examples seemed to be cherry-picked or based on authors' knowledge rather than from an extensive review of the literature.

We should have originally presented this article as a commentary article and not a research article. It is now noted in the abstract (line 22) and introduction (line 115) that this is a commentary article. As such, it is not intended to provide a comprehensive review of the evidence but rather to highlight opportunities for the use of tools, assessment, and strategic framing which could be used in complementary ways to apply a systems-based approach to support climate and health actions.

The article is not 'new research'; it is commentary proposing the use of systems thinking for health and climate adaptation. This has already been discussed in the literature in various contexts (see for example https://www.nature.com/articles/s41558-018-0102-4; https://www.pnas.org/content/116/17/8214; https://publichealthreviews.biomedcentral.com/track/pdf/10.1186/s40985-016-0032-5.pdf) yet none of these previous studies were mentioned.

We should have originally presented this article as a commentary article and not a research article. It is now noted in the abstract (line 22) and introduction (line 115) that this is a commentary article.

The Berry et al. 2018 and other relevant systems thinking references are now cited in lines 44-45.

What was disappointing was that the exact way that this 'new' approach could be implemented was not explained in sufficient detail for one to actually do it. Many statements made in the manuscript are sweeping and without substantiation or justification. For example, lines 157 - 159 what exactly does 'a fuller accounting of health impacts .... of climate policies is necessary.....' mean? Line 162, how can one manage proactively? Line 172, how can one value environmental health services? or identify the cost of actions needed for health systems to prevent, manage and respond...'? The reader is left to figure this out for themselves. Another example of the vagueness recurring in the manuscript is in line 196 - what are the important differences mentioned here? They are stated as being important yet are not elaborated upon. Line 322, how does one reduce the impacts of extreme weather on health? There are numerous examples in the manuscript where the reader is left wondering what the authors mean.

“Accounting” of health impacts is clarified in lines 218-224. The reasons that quantification is strategic is described in lines 221-222, 229-240. Examples in the literature of health impacts which have been quantified in the context of supporting the case for climate action are described in lines 253-269.  The types of methodologies are cited in lines 271-274. Differences in methodology of quantifying health in economic terms are noted in lines 276-280.

Health risks that are identified could be managed proactively through preventive measures which could include increasing awareness of the health workforce, among others, of climate change-health relationships; improved health information systems for integrated risk monitoring and early warning; and, greater financing of health systems to support these latter two activities. See lines 222-6.

The peat fire and palm oil example is interesting, but the specifics of the example make it difficult to see how this would be done for other climate-health problems. A more generic framework could help the reader 'see' what they would need to do to address any pressing problem.

This example is intended to illustrate how a systems-based understanding of climate change and health (i.e. the human drivers affecting their relationship including feedbacks, tradeoffs and unintended side effects) can help identify interventions such as improved protection of peatlands to benefit climate and health while at the same time reveal other issues (e.g. supporting farmers whose livelihoods depend on palm oil) that should be addressed in designing solutions. Such a systems-based understanding could apply to other human driven environmental changes affecting health such as land use change/deforestation and emerging infectious disease risk (Pulliam JRC, 2012) as well as deforestation and increased risk of diarrheal disease (Herrera D et al., 2016).

The Discussion section introduces text relating to the Green Climate Fund and seems to give the Fund specific guidance. Why is only this Fund mentioned? Other funds and donors support climate change and health adaptation. This was an odd discussion that did not bring the content of the manuscript together, instead it dealt with a different topic. This made the reader even more confused. 

The language has been revised to include The Green Climate Fund as one example of donors (lines 491-492) which we are suggesting should pay more attention to health in the climate and mitigation projects they sponsor. The Discussion section (lines 489-498, 529-532) makes summary recommendations on how the key players (e.g. countries, their partners, donors) developing climate-health actions and projects can play a role in using a systems-based approach as the basis for the use of existing tools, assessments and strategic framing in a complementary way – and, why this can increase the financial viability of climate-health actions and projects.

Specific comments

  1. The introduction should cover other articles that that considered systems thinking in climate change (and health) impacts as well as define what this manuscript sets out to achieve and how. There is no mention of methods - the authors should at least state this is an opinion piece and that the content is derived from select examples without a full review of the literature. In fact, such a review of the literature may be useful to provide more extensive examples of how the proposed systems-based understanding of climate change and health (using HIA and HiAP) and to then develop a framework or guidelines on exactly how to make use of this approach.
  2. A short methods section is needed - either as standalone or as part of the introduction.

We should have originally presented the manuscript as a commentary article and not a research article. It is now noted in the abstract (line 22) and introduction (line 115) that this is a commentary article. As such, it is not intended to provide a comprehensive review of the evidence but rather how the complementary use of existing impact assessment tools, quantitative assessment, and strategic framing can support the application of a systems-based approach to support climate and health actions. In this commentary, we are not applying a scientific method to a case study. The definition of systems thinking in this paper’s context of climate change-health is presented in lines 46-49. Panel 1 presents a summary of systems thinking in terms of understanding climate-health relationships to inform climate and health actions/interventions based on that understanding.

  1. Please explain or given examples for the term 'non-health/health influencing sectors' at first use.

This has been simplified to “health and other” or “health and non-health” in lines 37-38 and 116, respectively, in the context of cross-sector collaboration on climate and health.

  1. Line 345, what is Readiness resources? A reference is needed if this is a specific product.

“Readiness resources” has been removed in line 496 since these are specific to The Green Climate Fund project development process.

  1. Paragraph ending on line 292 does not relate well to the paragraph starting on line 294 - please create better flow.

A transition sentence has been added in lines 398-401.

  1. Please check the manuscript for very long sentences - sometimes five lines long - that make comprehension difficult.

Long sentences have been reduced for easier readability and comprehension.

  1. Line 317 - why the focus on private investors? Funds can come from elsewhere.

As noted in the section, the cost of climate adaptation can be too high for most governments in low and middle-income countries. Therefore, we are interested in bringing in the perspective of the private sector as a means to leverage public funding to address linked climate-health challenges. 

  1. Line 324 - please give concrete examples of the insurance being discussed here and where it is being applied in the world (preferably not another Asian example).

An example of the use of parametric insurance from Mauritania has been added in lines 437-445.

  1. In several paragraphs, there are no references. The authors should carefully check each sentence and make certain that the references are included to substantiate points made.

References have been added to text in the Discussion section (line 532) and in lines 511-519 regarding further research that is needed. In this commentary article, some suggestions on ways forward to inform climate actions with health benefits are based on country level experiences of both authors but may not have not been published.

Reviewer 3 Report

The promise of this paper's introduction - to lay out a roadmap for how to implement a systems-based approach to redress the health impacts of climate change is an interesting potential contribution which falls short in the narrative text that follows.

If you revise the manuscript, I'd suggest being more attentive to systems-based principles in Section 2. You are strong on tying systems principles back to your section on implementation, but the threads you began weaving in the introduction aren't adequately tied to the examples you raise in section 2. This would considerably strengthen the paper.

Re: HIA - would suggest also linking this back to the emerging discourse on climate change and health vulnerability and adaptation assessments (which is more en vogue than HIA in this space and redresses some of HIA's known shortcomings).

RE: Health in all policies - how can you tie critiques of HiaP (colonial, imperialistic) to the need to be humble and mindful of past precedents in this field, and of collaborators working actively on climate issues (many of whom have decades of experience relative to those who are just engageing with CC from a health perspective), etc? What does a systems perspective offer to implementing HiaP in a way that is inclusive and just?

Author Response

Thank you very much for reviewing the manuscript and providing constructive comments. Please see our responses (in bold) which are presented below the specific comments received (copied here for reference)

Reviewer 3

The promise of this paper's introduction - to lay out a roadmap for how to implement a systems-based approach to redress the health impacts of climate change is an interesting potential contribution which falls short in the narrative text that follows.

If you revise the manuscript, I'd suggest being more attentive to systems-based principles in Section 2. You are strong on tying systems principles back to your section on implementation, but the threads you began weaving in the introduction aren't adequately tied to the examples you raise in section 2. This would considerably strengthen the paper.

To tie back to systems-based principles as described in Section 1, we have revised and/or added language in Sections 2.1 (lines 147-149), Section 2.2 (lines 215-222), Section 2.3 (lines 291, 315-320), Section 2.4 (lines 350-354, 398-399).

Re: HIA - would suggest also linking this back to the emerging discourse on climate change and health vulnerability and adaptation assessments (which is more en vogue than HIA in this space and redresses some of HIA's known shortcomings).

We have added language on how the use of HIA can inform vulnerability and adaptation assessments in lines 339-344.

RE: Health in all policies - how can you tie critiques of HiaP (colonial, imperialistic) to the need to be humble and mindful of past precedents in this field, and of collaborators working actively on climate issues (many of whom have decades of experience relative to those who are just engageing with CC from a health perspective), etc? What does a systems perspective offer to implementing HiaP in a way that is inclusive and just?

Please see lines 315-320 including the benefits of a systems perspective to HiaP.

Reviewer 4 Report

  1. The emphasis on using a "systems approach" and on the value of HIA is welcome. But this emphasis, in itself, is not new and indeed the author has already covered some related ground in an earlier paper (reference 5). It would help if this manuscript showed more clearly how it builds on the earlier literature on systems thinking across climate-environment-health and why the particular examples were chosen for discussion.
  2. The abstract runs the risk of promising somewhat more than is actually delivered in raising expectations about delineating "strategic steps to put a systems approach into practice.." Assuming that these steps encompass "problem conceptualisation, design, to implementation and evaluation" (line 336) it would be helpful to see more about how the concrete examples presented can be analysed in terms if these steps. It might also help to return to Panel 1 in the Discussion in order to ensure that it is well-connected with the body of the text.
  3. Section 2.1 provides some interesting examples but only limited evidence that the practice of HIA is being mainstreamed or how a strategy is conceived and operationalised. Perhaps this section could be rethought by tabulating strengths and weaknesses of the examples chosen in terms of mainstreaming or other strategic elements? Also, it would be good to see more focus on the ASEAN experience (lines 146-153) in providing specific examples of good practice in mainstreaming HIA across sectors and linking national-regional strategies/coordinating policy actions. This ASEAN work is likely to be not so familiar to many readers and merits greater visibility.
  4. Section 2.2 also covers crucial points, for quantification of health (and other) impacts but it tends to ignore the practical difficulties associated with controversies in health economics. Although some literature is referenced (lines 193-198), no assessment is made on how best, for example, to value a life (especially the value of a statistical life) or how to compare the economic value of climate change mitigation and adaptation approaches between different demographic groups, societies etc. The problem is compounded because of the diversity of health outcomes in response to a diversity of climate change pathways and the concomitant objective to value other ecosystem services. While methodological solutions may not be yet entirely within reach, it would be good to see a more explicit recognition of the problems in quantification. Without better quantification more broadly, it is difficult to think about answering some really important questions, such as, do the immediate economic costs associated with climate action inhibit the potential influence of health co-benefits on development of mitigation policies? Although the example of air pollution is convincing in this regard, the field needs developing considerably.
  5. Section 2.3 on multiple objectives: In the discussion of "When resources are limited..." (line 211 and also line 68), the authors might consider using the language of the "triple win" for determining objectives to cover benefits for health, environment and equity. Also in section 2.3, no mention is made of the NDCs. Why not? This might help to make the case for mainstreaming health actions and their assessment.
  6. It seems surprising that the word "research" is mentioned only once in the text, yet I'm sure the authors would agree that there is a large research agenda to pursue for the systems approach. While there may not be space in this manuscript for extensive discussion of research priorities, it would be good to see acceptance of the point that research is also a crucial step in the systems approach strategy. The recent IJERPH paper on transdisciplinary research priorities (Ebi et al, including one of the present authors) could be cross-referenced in this regard.

Author Response

Thank you very much for reviewing the manuscript and providing constructive comments. Please see our responses (in bold) which are presented below the specific comments received (copied here for reference)

Reviewer 4

1. The emphasis on using a "systems approach" and on the value of HIA is welcome. But this emphasis, in itself, is not new and indeed the author has already covered some related ground in an earlier paper (reference 5). It would help if this manuscript showed more clearly how it builds on the earlier literature on systems thinking across climate-environment-health and why the particular examples were chosen for discussion.

We should have presented this article as a commentary article and not a research article. It is now noted in the abstract (line 22) and introduction (line 115) that this is a commentary article. As such, it is not intended to provide a comprehensive review of the evidence but rather to apply a systems-based approach to support climate and health co-benefits based actions through the complementary use of existing tools (e.g. HIA), quantitative assessment, and strategic framing.
HIA should be used to assess likely health impacts of policies or projects (lines 171-175); identify vulnerable groups (lines 171-181); and, inform interventions for climate action with health benefits (lines 182-197). To conduct HIA in the climate-health context, HIA should reflect the current understanding of climate-health relationships and relevant health indicators which are increasingly available and regularly updated. The reasons the HIA examples are chosen are explained in lines 172-73, lines 182-183, lines 192-195.

2. The abstract runs the risk of promising somewhat more than is actually delivered in raising expectations about delineating "strategic steps to put a systems approach into practice.." Assuming that these steps encompass "problem conceptualisation, design, to implementation and evaluation" (line 336) it would be helpful to see more about how the concrete examples presented can be analysed in terms if these steps. It might also help to return to Panel 1 in the Discussion in order to ensure that it is well-connected with the body of the text.

The abstract has been revised in lines 22-25 to clarify the aim of the paper. To tie back to systems based principles as described in Section 1, we have revised and/or added language in Sections 2.1 (lines 147-149), Section 2.2 (lines 220-222, 230-237), Section 2.3 (lines 291, 315-320), Section 2.4 (lines 350-354).

3. Section 2.1 provides some interesting examples but only limited evidence that the practice of HIA is being mainstreamed or how a strategy is conceived and operationalised. Perhaps this section could be rethought by tabulating strengths and weaknesses of the examples chosen in terms of mainstreaming or other strategic elements? Also, it would be good to see more focus on the ASEAN experience (lines 146-153) in providing specific examples of good practice in mainstreaming HIA across sectors and linking national-regional strategies/coordinating policy actions. This ASEAN work is likely to be not so familiar to many readers and merits greater visibility.

The HIA examples are described to illustrate how HIA can be applied to climate-health challenges to not only identify likely climate-driven health impacts and which groups are most affected but to also directly inform the design/scope of climate adaptation actions. More ASEAN experience on HIA is added in lines 207-210.

4. Section 2.2 also covers crucial points, for quantification of health (and other) impacts but it tends to ignore the practical difficulties associated with controversies in health economics. Although some literature is referenced (lines 193-198), no assessment is made on how best, for example, to value a life (especially the value of a statistical life) or how to compare the economic value of climate change mitigation and adaptation approaches between different demographic groups, societies etc. The problem is compounded because of the diversity of health outcomes in response to a diversity of climate change pathways and the concomitant objective to value other ecosystem services. While methodological solutions may not be yet entirely within reach, it would be good to see a more explicit recognition of the problems in quantification. Without better quantification more broadly, it is difficult to think about answering some really important questions, such as, do the immediate economic costs associated with climate action inhibit the potential influence of health co-benefits on development of mitigation policies? Although the example of air pollution is convincing in this regard, the field needs developing considerably.

We present the advantages of quantifying health impacts of climate change. This is reinforced in the Discussion Section lines 533-539.

While we agree that the available methods for quantifying health are controversial and that there is no one standard approach, we present how health quantification can help increase the viability of funding climate-health actions and attracting support (from the private sector) for such projects, as described in lines 430-432, 481-485. From our experience, the benefits of health quantification for use in cost-effectiveness analysis and cost-benefit analysis are not commonly appreciated among decisionmakers working on climate and health. Respectfully, in this commentary, we prefer not to wade into the controversy on how “best” to value a life. We note in lines 276-280 the specific factors which should be considered when evaluating the methodologies used in scientific estimates of economic cost savings and harms associated with climate-associated health impacts.

5. Section 2.3 on multiple objectives: In the discussion of "When resources are limited..." (line 211 and also line 68), the authors might consider using the language of the "triple win" for determining objectives to cover benefits for health, environment and equity. Also in section 2.3, no mention is made of the NDCs. Why not? This might help to make the case for mainstreaming health actions and their assessment.

NDCs are noted in lines 296-300.

6. It seems surprising that the word "research" is mentioned only once in the text, yet I'm sure the authors would agree that there is a large research agenda to pursue for the systems approach. While there may not be space in this manuscript for extensive discussion of research priorities, it would be good to see acceptance of the point that research is also a crucial step in the systems approach strategy. The recent IJERPH paper on transdisciplinary research priorities (Ebi et al, including one of the present authors) could be cross-referenced in this regard.

Research priorities are included in the Discussion Section (lines 505-519).

Round 2

Reviewer 2 Report

The authors have adequately addressed my comments.

Author Response

Thank you for your comments. Please see Introduction section, Lines 122-123 and in the Discussion section, Lines 588-593.

Reviewer 3 Report

Thanks to the authors for making revisions to this paper. While the changes made have somewhat improved the presentation of the paper, there are also changes that have detracted from the paper's significance. More importantly, the critical analysis is weak, lacks specificity, lacks clarity in terms of the potential of HIA, and in some cases, is not accurate (e.g. HIA can be both prospective or retrospective) or omits other important information (e.g. incorporation of not only quantitative health info, but also qualitative which is necesary to understanding pathways to vulnerability... these types of information are historically marginalized in HIA). Most importantly, this reviewer's concerns are not adequately addressed. Specifically, the biggest issue I have with this manuscript is that it is still not clear how HIA can be 'systematized' (your 'panel' has a series of 'business as usual' questions which seem to exemplify this???) to better address climate hcange and other complex problems. Your examples are interesting, but you haven't provided clear guidance or core literature on systems-thinking and tied it in a new or innovative way to give your reader an idea of how these tools would force us to 'do' HIA differently in the context of climate change and health equity.

Second, the issue of climate change and health vulnerability assessments (which could be argued are just a highly specific form of HIA) is still glossed over (see my original comments). This is where the bulk of the evidence of HIA (with a systems focus on climate change and equity) can be found, but it is still notably absent within your manuscript, despite profound national and international policy and practice recommendations.Perhaps an interrogation of how/why these are more/less rooted in systems-thinking than HIA would be helpful?

Finally, the point around HiaP (that health in all policies by its very nature is a colonizing concept) has not been addressed. I think the authors misinterpreted the comment. The notion here is that HIAs are often not even practiced by trained health professionals, and that forcing health into other fields can be viewed as health imperialism, especially if it is not mindful or appreciative of the decades of experience of those working within those fields. A true systems approach that is attentive to equity concerns needs to name this as such and provide clear, actionable detail on how to redress it.

Author Response

Thank you for your comments. Please see attachment. All responses are highlighted in yellow, following each comment (copied for reference).
